# GRAPH VIEW-CONSISTENT LEARNING NETWORK

## ABSTRACT

Recent years, methods based on neural networks have made great achievements in solving large and complex graph problems. However, high efficiency of these methods depends on large training and validation sets, while the acquisition of ground-truth labels is expensive and time-consuming. In this paper, a graph view-consistent learning network (GVCLN) is specially designed for the semi-supervised learning when the number of the labeled samples is very small. We fully exploit the neighborhood aggregation capability of GVCLN and use dual views to obtain different representations. Although the two views have different viewing angles, their observation objects are the same, so their observation representations need to be consistent. For view-consistent representations between two views, two loss functions are designed besides a supervised loss. The supervised loss uses the known labeled set, while a view-consistent loss is applied to the two views to obtain the consistent representation and a pseudo-label loss is designed by using the common high-confidence predictions. GVCLN with these loss functions can obtain the view-consistent representations of the original feature. We also find that preprocessing the node features with specific filter before training is good for subsequent classification tasks. Related experiments have been done on the three citation network datasets of Cora, Citeseer, and PubMed. On several node classification tasks, GVCLN achieves state-of-the-art performance.

## 1 INTRODUCTION

Convolutional neural networks (CNNs) (Krizhevsky et al., 2012) performed outstandingly in solving problems such as image classification (Rawat & Wang, 2017), semantic segmentation (Kampffmeyer et al., 2016) and machine translation (Cho et al., 2014) etc. This is because CNNs can effectively reuse the convolution kernel and use the given input to train optimal parameters. The original data mentioned in above problems all have a grid-like data structure, that is, Euclidean spatial data. In reality, there are also lots of non-Euclidean spatial data, such as social networks, telecommunication networks, biological networks, and brain connection structures, etc. These data are usually represented in the form of graphs, where every node in the graph represents a single individual. Graph problems can be roughly divided into there direction: link prediction (Zhang & Chen, 2018), graph classification (Zhang et al., 2018a) and node classification (Kipf & Welling, 2016). In this paper, we focus on semi-supervised node classification when the label rate is very low.

Many new methods have been proposed to generalize the convolution operation to process graph structure data on arbitrary graphs for node classification. These methods can be divided into spatial convolution and spectral convolution methods (Zhang et al., 2018b). For spatial methods, they directly define graph convolution by designing certain operations on node's neighbors. For example, Duvenaud et al. (2015) propose a convolutional neural network that can directly operate on graph data, which can provide an end-to-end feature learning method; Atwood & Towsley (2016) propose a fusion convolutional neural network (DCNNS), which introduces the graph fusion method to incorporate the context information of the nodes in the graph node classification; The Graph Attention Network (GATs) (Veličković et al., 2017) introduces the attention mechanism into the graph data processing to construct the attention layer for semi-supervised learning. The spectral method generally defines the graph convolution operation on spectral representation of graph. For example, Bruna et al. (2013) propose that graph convolution can be defined in the Fourier domain based on the eigenvalue decomposition of the graph Laplacian matrix; Defferrard et al. (2016) propose to use the Chebyshev expansion of the graph Laplacian to approximate the spectral domain filtering, which can

avoid high computational complexity brought by eigenvalue decomposition; Kipf & Welling (2016) propose a simpler Graph Convolutional Networks (GCNs) for semi-supervised learning, which can achieve higher classification accuracy by using a simple two-layer networks. However, large training and validation sets are required in these methods to complish effect classification task, while obtaining true labels is time-consuming, laborious and costly. And they send original graph node features directly into networks for training, however, there are lots of redundant information in the original features of the nodes.

In order to train an efficient model with only a few label nodes and even without validation, we put forward our own method: graph view-consistent learning network (GVCLN), which constructs a node classification network based on the consistency between two views. First, we independently train two-view encoders (can be different) to obtain two representations of every node. The function of the viewers is converting high-dimensional node features into low-dimensional embeddings (Zhu et al., 2020). The clustering hypothesis (Vandenberg & Matthias, 1977) show that examples in the same cluster are more likely to have the same label. According to this hypothesis, the decision boundary should try to pass through the place where the data is relatively sparse, so as to avoid dividing the data points in dense clusters on both sides of the decision boundary. Although the two views have different viewing angles, their observation objects are the same, so their observation results should be consistent. Therefore, the features encoded by the two viewers should make the decision boundary pass through the place where the date is sparse, that is, there should be consistency between the two views. Then, we design three loss functions, namely, supervised loss function, consistency loss function, and pseudo-label loss function. The supervised loss uses the known labeled set, while a view-consistent loss is applied to the two views to obtain the consistent representation and a pseudo-label loss is designed by using the common high-confidence predictions as pseudo label. GVCLN with these loss functions can obtain the view-consistent representation of the original feature. Our contributions are summarized as follows:

- We propose a graph view-consistent learning framework for semi-supervised node classification, which fully demonstrates the theoretical structure of graph view-consistency.

- We design GVCLN to successfully tackle label insufficiency in semi-supervised learning.

- We demonstrate the high efficacy and efficiency of the proposed methods on various semi-supervised node classification tasks.

## 2 RELATED WORK

### 2.1 NOTATIONS

A graph contains two parts: node and edge. Each node represents an individual, which can be a paper or a person, etc. The edge indicates a connection between two nodes. If the edge connecting two nodes in the graph is directional, it is directed graph, otherwise it is undirected graph. A simple and connected undirected graph can be written as $G = (V, E)$, where $V$ is the node set and $E$ is the set of edges. $n = |V|$ represents the number of all nodes in $G$. Considering that the node itself has a great influence on the graph structure, the graph used in the calculation of the network is generally a self-loop graph, namely $\tilde{G} = \left(V, \tilde{E}\right)$, which attaches a self-loop to each node in $G$. $A$ denotes the adjacency matrix and $D$ is the diagonal degree matrix. Therefore, the adjacency matrix and diagonal degree matrix of $\tilde{G}$ are defined as $\tilde{A} = A + I$ and $\tilde{D}$, respectively. $I$ indicates the identity matrix. The node feature matrix is $X \in \mathbb{R}^{n \times d}$, in which, each node $i$ is associated with a $d$-dimensional feature vector $\boldsymbol{x}_i$. The normalized graph Laplacian matrix is defined as $L = I - D^{-1/2}AD^{-1/2}$, which is a symmetric positive semidefinite matrix with eigendecomposition $U\Lambda U^\top$, where $\Lambda$ is a diagonal matrix of the eigenvalues of $L$, and $U \in \mathbb{R}^{n \times n}$ is a unitary matrix that consists of the eigenvectors of $L$. The graph convolution operation between signal $\boldsymbol{x}$ and filter $g_\gamma(\Lambda) = \text{diag}(\gamma)$ is defined as $g_\gamma(L) * \boldsymbol{x} = U g_\gamma(\Lambda) U^\top \boldsymbol{x}$, where the parameter $\gamma \in \mathbb{R}^n$ corresponds to a vector of spectral filter coefficients.

## 2.2 FEATURE AGGREGATION

Graph neural Networks (GNNs) follow a neighborhood feature aggregation scheme. Many GNNs use different aggregation approaches. Here, we introduce two most classic feature aggregation ways.

Graph convolutional network (GCN) (Kipf & Welling, 2016) has outstanding effects in solving the problem of semi-supervised node classification. Kipf & Welling (2016) define the graph convolutional layer for feature aggregation in GCN. Symmetrically normalized operation is used on the adjacency matrix of $\tilde{G}$, i.e., $\hat{A} = \tilde{D}^{-1/2}\tilde{A}\tilde{D}^{-1/2}$. The graph convolutional layer is defined by,

$$H^{l+1} = \sigma(\hat{A}H^l\Theta), \tag{1}$$

where $H^l$ is the feature of the $l$-th layer, $\Theta$ is the trainable weight matrix of the layer, and $\sigma(\cdot)$ is a nonlinear activation function, e.g., $\text{ReLU}(\cdot) = \max(0, \cdot)$.

Veličković et al. (2017) present graph attention network (GAT), novel neural network architectures that operate on graph-structured data, leveraging masked self-attentional layers to address the shortcomings of prior methods based on graph convolutions or their approximations. By giving different attention weights $\alpha_{ij}$ to different neighbor nodes, each node-$i$ are able to aggregate their neighborhoods features to update their own features. The graph attention layer is defined by,

$$\boldsymbol{h}_i^{l+1} = \sigma\Big(\sum_{j \in N_i} \alpha_{ij} W \boldsymbol{h}_j^l\Big), \tag{2}$$

where $\boldsymbol{h}_i$ represents latent embedding feature at node $i$, $W$ is a linear transformations weight matrix, and $\alpha_{ij}$ are normalized attention coefficients.

## 3 MODEL ARCHITECTURE

In this section, we first introduce the overall abstract structure of GVCLN, and then use graph convolutional layer and graph attention layer to build a specific GVCLN model to achieve view-consistent learning for node classification.

## 3.1 MODEL ARCHITECTURE OVERVIEW

A general understanding of our GVCLN is shown in Fig. 1. In GVCLN, feature $X$ and the graph $A$ are encoded by two different viewers at the same time, namely Viewer 1 and Viewer 2, the two viewers aggregate features with three-head graph layers. Viewer 1 uses the graph convolution layer and Viewer 2 is the graph attention layer. The three-head representations, $H_1^{(v)}, H_2^{(v)}, H_3^{(v)}, v \in \{1, 2\}$, are of concatenation and perform dropout. After dropout, the two-view latent features, $Z^{(1)}$ and $Z^{(2)}$, pass through a same non-linear graph convolutional layer to obtain the prediction, $P^{(1)}$ and $P^{(2)}$. A view-consistency loss function is made for the dual-view output consistently.

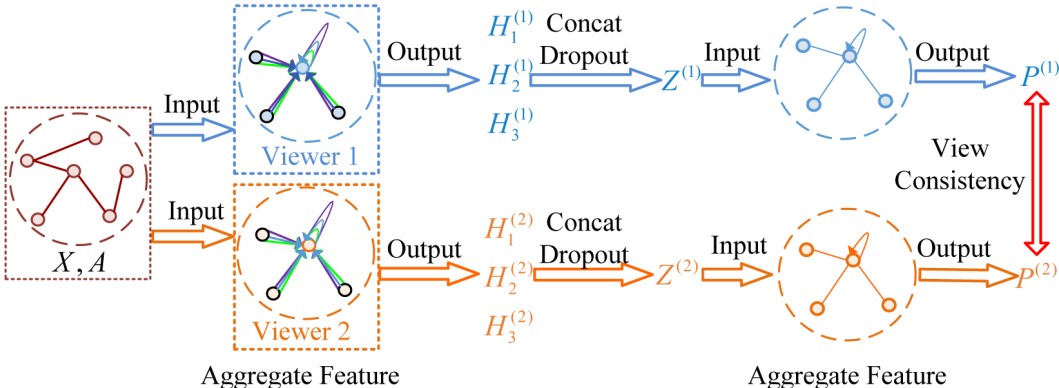

Figure 1: GVCLN architecture

### 3.2 VIEW-CONSISTENCY LEARNING

We consider applying view-consistent learning to the view 1 and view 2 in Figure 1. Inspired by contrastive learning, Chen et al. (2020) inputs the original sample and the sample after data augmentation (flipping or cropping, etc.) into two same views respectively. Since data augmentation does not change the label information of sample, the output representations of two views are similar. By maximizing the consistency of the two views, better results can be achieved. In this paper, we do not do data augmentation to the data, but used two different viewers. We believe that the same data has the similar output through different viewers and classifiers. In Figure 1, there is a consistency loss function between the view 1 and the view 2 to realize view-consistent learning of two views. The soft cross entropy function is employed to construct the consistency loss function:

$$\ell_{\text{view}} = -\frac{1}{n} \sum_{i=1}^{n} \sum_{j=1}^{k} p_{ij}^{(1)} \ln p_{ij}^{(2)}. \tag{3}$$

where $k$ is the number of class, $P^{(1)} = [p_{ij}^{(1)}]$ and $P^{(2)} = [p_{ij}^{(2)}]$ are the predictions of the two viewers for node-$i$, respectively.

The concept of co-training was proposed by (Blum & Mitchell, 1998), co-training assumes that each data point has two views, and each view is sufficient for learning an effective model. Co-training learns separately on two views with labeled data samples to obtain two different learners and then uses the unlabeled data samples in the respective views of the two learners to make predictions. The basic learning process of collaborative training is: Each learner selects the data sample with the top high-confidence predictions in unlabeled data and adds it to the labeled data sample set of another learner for training. This process is repeated until a specific stopping condition is met. Co-training needs to meet the following two conditions:

- Views are sufficient and redundant, that is, for each view, if enough labeled data is given, each view can learn a learner with good performance.
- Conditionally independent, that is, the label information of each view is conditionally independent of the label information of another view.

Our view-consistent learning method is similar to co-training, which requires pseudo-labels for training. The selected pseudo-label data cannot be completely clean (the predicted label is exactly the same as the real label), but contains noise (the situation where the predicted label is inconsistent with the real label). Due to the powerful representation capabilities of deep neural networks, even noisy labels can be fitted by deep neural networks. Based on (Han et al., 2018), which can train deep neural networks with extremely noisy labels, we propose our own training strategy. We use labeled samples to train two independent learners, namely view 1 and view 2. When the two-view networks are well trained, according to their predictions, the predicted labels of some nodes with the same prediction representations and common high-confidence prediction are selected as pseudo labels. Then, pseudo label set is added to the original labeled set for training. This is repeated until the stop condition is reached.

We believe that pseudo labels have similar importance as real labels, but the number of pseudo-labels taken each time can be controlled by the loss function value and the accuracy of the verification set, and different numbers of pseudo-labels can be given to data sets with different label rates.

GVCLN selects nodes with high confidence and with same prediction results in the two views as pseudo labels. The pseudo labels are voted with high-confidence predictions of the two viewers taken from unlabeled nodes.

The pseudo-label loss has the same effect as consistency loss, that is: making the two views more and more consistent. Meanwhile, supervised loss has the function of fine-tuning and controlling the direction of network optimization.

We choose cross entropy as a loss function to calculate the loss for both the labeled samples and the pseudo labels:

$$\text{CE}_i = -\sum_{j=1}^{k} y_{ij} \ln p_{ij}, \tag{4}$$

where $y_{ij}$ denotes the ground-truth label, i.e., the $i$-th node belongs to the $j$-th class, $p_{ij}$ is the network prediction probability of the $i$-th label.

Thus, the supervised loss function of labeled samples is:

$$\ell_{\text{sup}} = -\frac{1}{s} \sum_{i=1}^{s} \sum_{j=1}^{k} y_{ij} \ln p_{ij}. \tag{5}$$

where $s$ denotes the number of labeled nodes,

For pseudo labels, we also use cross entropy to calculate the loss function:

$$\ell_{\text{pseudo}} = -\frac{1}{t} \sum_{m=1}^{t} \sum_{i=1}^{k} y'_{ij} \ln p_{ij}. \tag{6}$$

where $y'_{ij}$ is the pseudo label and $t$ denote the number of nodes of the pseudo labels.

### 3.3 ALGORITHM

Then, the total loss of GCVLN is given by:

$$\ell_{\text{total}} = \ell_{\text{sup}}^{(1)} + \ell_{\text{sup}}^{(2)} + \beta \ell_{\text{view}} + \ell_{\text{pseudo}}, \tag{7}$$

where $\beta$ is a trade-off parameter, which is used to weigh the label loss function and the consistency loss function.

We first pre-train the GVCLN with $\ell_{\text{view}}$ and $\ell_{\text{sup}}$, and second use the total loss function, $\ell_{\text{total}}$. The specific process of GVCLN can be seen in Algorithm 1.

---

**Algorithm 1** The GCVLN algorithm.

1: **Input:** $X, A, \text{epoch}_{\max}, \text{epoch}_{\text{pre}}, \beta$.
2: **Output:** $P^{(1)}$ and $P^{(2)}$.
3: **Initialize:** Model initialization.
4: $X = \hat{A}^m X$
5: **for** $\text{epoch} \in [1, \text{epoch}_{\max}]$ **do**
6:     **if** $\text{epoch} < \text{epoch}_{\text{pre}}$ **then**
7:         $\ell = \ell_{\text{sup}}^{(1)} + \ell_{\text{sup}}^{(2)} + \beta \ell_{\text{view}}$ .
8:         Do optimization.
9:     **else**
10:         $\ell_{\text{total}} = \ell_{\text{sup}}^{(1)} + \ell_{\text{sup}}^{(2)} + \beta \ell_{\text{view}} + \ell_{\text{pseudo}}$ .
11:         Do optimization.
12:     **end if**
13: **end for**

---

In this paper, we use a renormalization filter (Li et al., 2019) to preprocess features as $X \leftarrow \hat{A}^m X$. The filter strength is tuned by the filter parameter $m$. When the label rate is low, $m$ is increased, and when the label rate is high, $m$ is reduced. By this operation, label efficient can be achieved.

## 4 EXPERIMENT

In this section, we use GVCLN to carry out related experiments and analyze the results in detail. The datasets include three citation network of Cora, Citeseer and PubMed. We conducted experiments with different label rates and without verification set. The division of the experimental dataset is the same as Luan et al. (2019). Classification accuracy of the test set is used as metric to evaluate the quality of model.

Table 1: Datasets statistics.

| Dataset | Vertices | Edges | Classes | Features |
|---------|----------|-------|---------|----------|
| Cora | 2708 | 5429 | 7 | 1433 |
| Citeseer | 3327 | 4732 | 6 | 3703 |
| PubMed | 19717 | 44338 | 3 | 500 |

## 4.1 DATASETS

We use three citation datasets: Cora, Citeseer and PubMed, which are widely used in graph problems. The specific information of the three data sets can be found in Table 1.

**Cora:** The Cora dataset contains 2708 nodes, which represent 2708 scientific documents and they can be divided into 7 categories. The citation network has 5,429 connection references, so the resulting graph contains 5,429 edges. The node feature is represented by a word vector containing 1433 independent and unique words, that is, the node feature is represented by a 1433-dimensional vector.

**Citeseer:** The Citeseer dataset contains 3327 nodes, which can be divided into 6 categories and each node represents one document. The citation network contains 4732 citation relationships. The feature of each document is represented by a word vector containing 3703 independent and unique words, that is, the feature of a node is represented by a 3703-dimensional vector.

**PubMed:** The PubMed dataset contains 19717 scientific papers, which can be divided into three categories. The citation network contains 44,338 citation relationships. The feature of each paper is represented by a word vector containing 500 independent unique words, and the feature of a node is represented by a 500-dimensional vector.

## 4.2 SETTING

GVCLN use graph convolutional layer and graph attention layer to build two viewers. The dimension of the hidden layer of Viewer 1 is set to 16, the drop rate is set to 0.2, the first large layer uses three-head graph convolutional layer to concatenate together, the second large layer uses only one graph convolutional layer; The dimension of the hidden layer of Viewer 2 is set to 8, the drop rate is set to 0.6, the first large layer uses three-head graph attention layer to concatenate together, and the second large layer uses only one graph convolutional layer. Other specific parameters are given in Appendix. Since the quality of the training set sampled each time is different, when the sampled training set node is located in the cluster center, the training and testing results are better. When the sampled training set node is located at the classification boundary, it will cause the training to be biased and the effect is poor, so the experimental results of this paper are the average results of ten tests. In the experiment, when the label rate is low, the filtering strength is increased and the number of pseudo labels is increased. As the label rate increases, the filtering strength and the number of pseudo labels are gradually reduced. At the same time, when the number of real label increases, the number of pseudo labels should be appropriately reduced.

## 4.3 RESULTS

In our paper, the framework of view-consistent learning is applied to the problem of semi-supervised graph node classification and combined with graph feature filtering and pseudo-label learning, so that the GVCLN performs outstandingly when dealing with low label rate problems and without validation. The division of the experimental dataset is the same as Luan et al. (2019). Classification accuracy of the test set is used as metric to evaluate the quality of model. The test results without validation can be seen in Table 2. Then, we make a specific analysis of the test results of the three datasets.

Our baseline is Stronger Multi-scale Deep Graph Convolutional Networks (Luan et al., 2019), which generalize spectral graph convolution and deep GCN in block Krylov subspace forms and devise two architectures, both with the potential to be scaled deeper but each making use of the multi-scale information differently. Luan et al. (2019) designed three sub-models: Linear Snowball, Snowball and Truncated Krylov, all of which can efficiently classify graph nodes. We used the same experimental

setup as Luan et al. (2019), and conducted training and testing without a validation set and with different label rates. Our method GVCLN can surpass Luan et al. (2019) under different label rates. In the appendix, we give a list of parameter settings for GVCLN and Luan et al. (2019). We also compare against other methods, including label propagation using ParWalks (LP) (Wu et al., 2012), Chebyshev networks (Cheby) (Defferrard et al., 2016), Co-training (Li et al., 2018), Self-training (Li et al., 2018), Multi-stage self-supervised (M3S) training (Sun et al., 2019b), graph convolutional networks (GCN) (Kipf & Welling, 2016), GCN with sparse virtual adversarial training (GCN-SVAT) (Sun et al., 2019a) and GCN with dense virtual adversarial training (GCN-DVAT) (Sun et al., 2019a).

The second, third and fourth column of Table 2 show the accuracy of our proposed method GVCLN and other existing methods under different label rates. It can be seen that GVCLN can reach the highest level on three citation datasets. For Citeseer, when the labeling rate is 0.5%, the GVCLN test accuracy rate is nearly 6.4% higher than the highest level of Truncated Krylov.

Table 2: Accuracy Without Validation

| Dataset | Cora | | | | | | Citeseer | | | | | | PubMed | |
|---|---|---|---|---|---|---|---|---|---|---|---|---|---|---|
| Label rate | 0.5% | 1% | 2% | 3% | 4% | 5% | 0.5% | 1% | 2% | 3% | 4% | 5% | 0.03% | 0.05% |
| LP | 56.4 | 62.3 | 65.4 | 67.5 | 69.0 | 70.2 | 34.8 | 40.2 | 43.6 | 45.3 | 46.4 | 47.3 | 61.4 | 66.4 |
| Cheby | 38.0 | 52.0 | 62.4 | 70.8 | 74.1 | 77.6 | 31.7 | 42.8 | 59.9 | 66.2 | 68.3 | 69.3 | 40.4 | 47.3 |
| Co-training | 56.6 | 66.4 | 73.5 | 75.9 | 78.9 | 80.8 | 47.3 | 55.7 | 62.1 | 62.5 | 64.5 | 65.5 | 62.2 | 68.3 |
| Self-training | 53.7 | 66.1 | 73.8 | 77.2 | 79.4 | 80.0 | 43.3 | 58.1 | 68.2 | 69.8 | 70.4 | 71.0 | 51.9 | 58.7 |
| M3S | 61.5 | 67.2 | 75.6 | 77.8 | 78.0 | - | 56.1 | 62.1 | 66.4 | 70.3 | 70.5 | - | 59.2 | 64.4 |
| GCN | 42.6 | 56.9 | 67.8 | 74.9 | 77.6 | 79.3 | 33.4 | 46.5 | 62.6 | 66.9 | 68.7 | 69.6 | 46.4 | 49.7 |
| GCN-SVAT | 43.6 | 53.9 | 71.4 | 75.6 | 78.3 | 78.5 | 47.0 | 52.4 | 65.8 | 68.6 | 69.5 | 70.7 | 52.1 | 56.9 |
| GCN-DVAT | 49.0 | 61.8 | 71.9 | 75.9 | 78.4 | 78.6 | 51.5 | 58.5 | 67.4 | 69.2 | 70.8 | 71.3 | 53.3 | 58.6 |
| Linear Snowball | 69.5 | 74.1 | 79.4 | 80.4 | 81.3 | 82.2 | 56.8 | 65.4 | 68.8 | 71.0 | 72.2 | 72.2 | 64.1 | 69.5 |
| Snowball | 67.2 | 73.5 | 78.5 | 80.0 | 81.5 | 81.8 | 56.4 | 65.0 | 69.5 | 71.1 | 72.3 | 72.8 | 62.9 | 68.3 |
| Truncated Krylov | 73.0 | 75.5 | 80.3 | 81.5 | 82.5 | 83.4 | 59.6 | 66.0 | 70.2 | 71.8 | 72.4 | 72.2 | 69.1 | 71.8 |
| **GVCLN** | **73.5** | **76.4** | **80.9** | **82.7** | **83.2** | **84.4** | **65.9** | **69.4** | **71.0** | **72.3** | **72.4** | **72.9** | **70.5** | **72.3** |

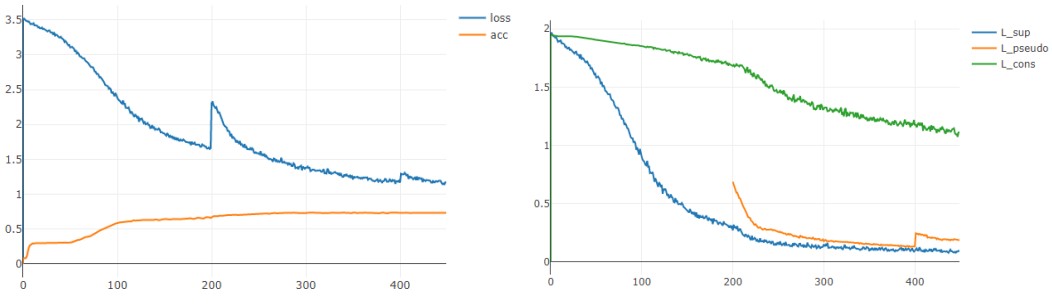

Figure 2: accuracy and loss of Cora 0.5%          Figure 3: all loss of Cora 0.5%

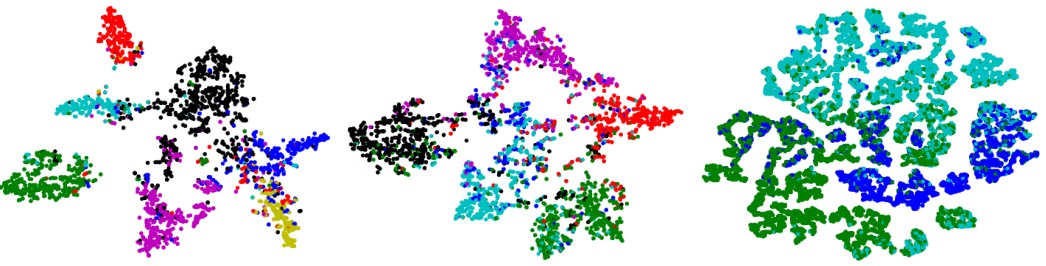

Figure 4: Cora with 5.6%          Figure 5: Citeseer with 3.6%          Figure 6: PubMed with 0.3%

Fig. 2 and Fig. 3 show the accuracy curve and loss function curve of the Cora dataset when the label rate is 0.5%. It can be seen that adding a pseudo-label when the epoch of training is 200 is helpful for the accuracy and loss function decline.

Fig. 4 is a t-SNE visualization diagram of the output results when each category contains 20 labeled samples, that is, the label rate is 5.6%. It can be seen from the visualization that the classification effect is still relatively good. The seven categories of the Cora data set finally appear to be clustered. Figure 5 and Figure 6 are the t-SNE visualization diagrams of the Citeseer data set and PubMed data set, respectively.

The renormalization filter with filter strength $m$=20 is used in the GVCLN model to preprocess the node features, which plays an important role in achieving label efficiency. In the experiment, we performed t-SNE visualization analysis on the original node features and filtered node features respectively, and the visualization results can be seen in Figure 7 and Figure 8. It can be clearly seen that all nodes in the original node feature visualization result are clustered together, and the node features after filtering are scattered and show certain signs of clustering.

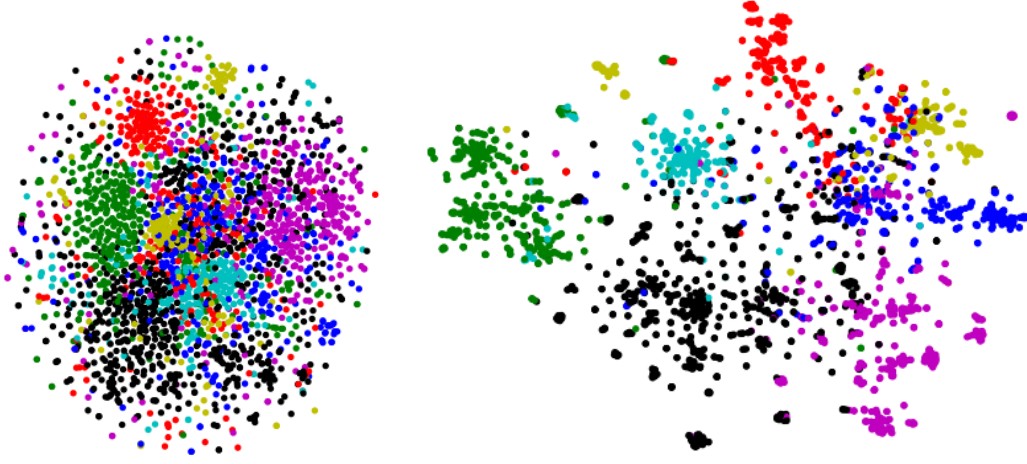

Figure 7: t-SNE of Cora raw features      Figure 8: t-SNE of Cora features with filtering

## 5 CONCLUSIONS

We propose a new model, GVCLN, to solve the node classification problem in the case of low label rate. GVCLN adopts a dual-view structure to view-consistent learning. For the two viewers, we use the graph convolutional layer and the graph attention layer, respectively, and finally pass through a non-linear graph convolutional layer. Because the graph convolutional layer of viewer 1 is relatively simple, it can quickly learn to obtain node representations, and then increases the representation ability of graph attention layer by supervised the consistency loss function in the learning. In contrast, the graph attention layer of viewer 2 is more complicated, it can prevent the addition of pseudo-labels from making the GVCLN unstable, and prevent the pseudo-label errors amplify step by step. Thus, We propose a view-consistency learning method, and carry out relevant practices on the task of graph node classification. Good results can be obtained on the three citation datasets on all label rate and without validation.

We also found that when the label rate is very low, there may be only a few labeled nodes. If these few labeled nodes are well represented (can support the entire dataset), the classification effect is particularly significant. On the contrary, when they are not enough to support the entire dataset, the classification effect will be worse. The direction of our future work will shift from a semi-supervised low label rate to unsupervised direction.

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
