# OpenReview forum: "Graph View-Consistent Learning Network"
_ICLR.cc/2021/Conference — Reject_

### Official Review · AnonReviewer2 · 2020-10-26

**Rating:** 3
**Confidence:** 4

**Review:**

[Summary]
In this paper, a graph view-consistent learning framework (GVCLN) is proposed. Specifically, two view learners are used to give predictions for the input. Then, a consistency loss is employed to force the two viewers giving the same predictions. Moreover, a co-training scheme is proposed to alleviate the label sparsity problem.

[Pros]
+ This paper is easy to follow.

[Cons]
- The novelty of this work is limited. It seems that this work is a simple combination of [1] and [2], with slight modification. Also, the authors are suggested to include more baselines, especially augmentation-based methods, e.g., [3].
- Three studied datasets are of small scales, which are well know to have unstable results.
- It is not clear how to select high-confidence pseudo labels. More experiments, e.g., parameter sensitivity analysis wrt the confidence threshold, ablation studies of the two loss, are needed.

[Evaluation]
Overall this paper presents a simple yet effective framework to semi-supervised node classification. However, the novelty of this work is limited and more experiments are necessary.

[Ref]
[1] A Simple Framework for Contrastive Learning of Visual Representations, in ICML, 2020

[2] Multi-Stage Self-Supervised Learning for Graph Convolutional Networks on Graphs with Few Labeled Nodes, in AAAI, 2020

[3] NodeAug: Semi-Supervised Node Classification with Data Augmentation, in KDD, 2020

---

> ### Author Response · Authors · 2020-11-15
> **Response to Reviewer 2**
>
> First of all, thank you for your valuable comments on our paper. We have made a detailed description of how to select pseudo labels in the appendix of the paper. The appendix is placed in the additional materials of the paper, not at the end of the paper. This is our negligence. We accept your comments humbly.

---

### Official Review · AnonReviewer3 · 2020-10-27
**This paper tries to address the limited training data problem by exploiting the consistency between different views of the graph data. However, important details and justification are missing.**

**Rating:** 3
**Confidence:** 4

**Review:**

This paper tries to address the limited training data problem by exploiting the consistency between different views of the graph data. However, important details and justification are missing. The major problems are:

(1)	The proposed model lacks detailed explanation and justification. How do you generate the view for the graph from its features? How and why do you obtain three head representations using graph convolution or graph attention? Why using convolution and attention for the two views respectively? How do you do dropout? Why using the same graph convolution layer later for the two views? After seven lines of introduction of the model, the paper is focused on training, and leave all the above questions behind.
(2)	For training, how do you judge ˋˋhigh confidence predictions’’ for generating pseudo-labels? What do you mean by ˋˋthe same prediction representations’’, ˋˋverification set’’ and ˋˋlabel rates’’? Moreover, with limited training data, how do you obtain ˋˋwell trained’’ two-view networks? What is the ˋˋstop condition’’, simply max epochs?
(3)	The authors give some settings of parameters and say that the other parameters are specified in the appendix. However, after references there is no appendix.

To summarize, without justification, the proposed model is not convincing. Without details of the model and implementation, this work is difficult to reproduce.

---

> ### Author Response · Authors · 2020-11-15
> **Response to Reviewer 3**
>
> First of all, thank you for your valuable comments for our paper. Due to the length of the article, we did not give a detailed explanation of the specific theory. In the appendix, we have made a detailed description of the parameter setting and the selection of pseudo-labels. The appendix is in the additional materials submitted, but not directly added to the end of the paper. This is an oversight of our work. For the lack of theoretical proofs you mentioned, we will pay more attention to the improvement of the paper in the future. Thank you for your comments.

---

### Official Review · AnonReviewer1 · 2020-10-29
**A multi-view learning approach for inferring graph representation with incremental improvement in performance**

**Rating:** 4
**Confidence:** 5

**Review:**

This paper adopts a multi-view learning approach for graph representation learning where some labels are assumed to be available. It uses graph convolution network (GCN) and graph attention network (GAT) to create two different views of the same graph and then define a loss function to force the output due to the two views to be consistent. The low label rate scenario is considered and pseudo labels are created to define an additional loss function to better enforce consistency. Three datasets are used for performance evaluation.

Pros:
- The problem addressed is an important one.
- The problem formulation is a reasonable one.
- The paper is clearly presented in general.

Cons:
- The view-consistency idea is good but not particularly new. The two graph “views” are based on existing graph embedding methods. So, I consider the originality of this work is limited.
- Only incremental improvement in performance is demonstrated in the experimental results.
- The graphs being evaluated are not particularly large.

Comments:
- In Eq 7, the two terms for supervised loss l_sup^(1) and l_sup^(2) are not clearly defined.

Qn:
- In the conclusion, there is some discussion about how pseudo labels contribute. It will be interesting to see how crucial the pseudo label term contributes to the overall the performance.
- How will the proposed method be compared with some contrastive learning in terms of performance?

---

> ### Author Response · Authors · 2020-11-15
> **Response to Reviewer 1**
>
> First of all, thank you for your valuable comments on our paper. Regarding your question of how pseudo-labels work, we used the two loss functions and accuracy graphs in the experiment to give a proof, which shows that adding pseudo-labels can indeed improve the performance of the network. As for the comparison with the contrastive learning model, since the baseline used in our paper is "Break the ceiling: Stronger multi-scale deep graph convolutional networks.", there is no need to compare with the contrastive learning. We will humbly adopt the advantages and disadvantages you put forward.

---

### Official Review · AnonReviewer4 · 2020-10-30
**Lack of innovation and lack of clear architecture analysis**

**Rating:** 4
**Confidence:** 4

**Review:**

This paper proposes a view-consistent framework to address the issues of expensive labels. In particular, this work first uses graph neural networks and graph attention networks to construct two different latent features of the same data. Then, it uses the same classification neural networks to produce the node classification outcomes. Finally, it uses the classification outcome to construct a so-called "view loss". In addition, it uses an incremental strategy to gradually included pseudo labels until some termination conditions are satisfied.

Overall, the paper is easy to understood. However, I think the paper can be improved in each sections:
[Introduction & Related work]
The authors can better organize their presentation on the development and understanding of Graph Neural Networks. At the current stage, these content does not seem to connect to the current development of GNN.

[Model architecture]
3.1
1, can the authors explain the reason of using three-head representation? Also, why do the authors use the same non-linear graph convolution layers? Is it because the feature is different already? Can the authors specify the detail settings on this graph convolution layers? I did not find it in other places?

3.2
2, I can roughly understand the reason of introducing the contrastive learning and co-training. However, maybe the authors should put of the content in the related work part and emphasise the difference of view-consistent algorithm from these two methods. Plus, I did not find experiments that uses the data augmentation method (instead of the view-consistent method). I can see there is contrast learning comparison, however, the other settings of constrast learning may be different?
3, the inclusion of pseudo labels are not well explained in this section. I was expecting to see more systematic analysis on this procedure, however, the current version can not fully convince me on this procedure.
4, small issues: view 1 and viewer 1 are both used in the paper and they should be consistent. Eq. (5) and Eq. (6) can be well formatted to save more space for presentation.

[Experiments]
5, I am expecting to see the implementation code for at least the neural network specification in the main paper or supplementary material. However, they are missing.

---

> ### Author Response · Authors · 2020-11-15
> **Response to Reviewer 4**
>
> First of all, thank you for your valuable comments on every part of our paper, and we will pay more attention to it in future paper writing. [Introduction & Related work] We introduced some existing papers based on spectral methods and non-spectral methods, and introduced some methods related to our papers. But there is really a lack of introduction to the development of the entire graph neural network. [Model architecture] Our model uses a three-headed representation. This is because we have found in experiments that using three-headed representation and randomly discarding can make the network retain more useful representation information and prevent over-fitting. The difference between view 1 and view 2 is mainly reflected in the first half, so the final output layer, we use clique convolution layer. For the setting of the volume layer, we have a detailed introduction in the "Settings" of the experiment. We did not upload our code in time, this is our negligence, but we will publish our code as soon as the paper is received.

---

### Official Review · AnonReviewer5 · 2020-11-05
**A multi-view contrastive learning framework for node classification with fewer labels.**

**Rating:** 5
**Confidence:** 5

**Review:**

Advantage:

The paper concentrated on an important perspective of graph learning: to utilize a small number of labels for large-scale graph learning. The framework is well demonstrated and the paper is easy to follow.

Weakness:

1. Novelty: My main concern of the paper is about the paper's novelty. There are already some works having similar multi-view contrastive learning frameworks, such as [1]. `Besides, The learning loss of the proposed method is similar to the self-training loss in SimCLR v2 [2].

2. Experiment: The experiment results are not sufficient. The author claimed the proposed method is efficient for graph classification on large training datasets. But the experiment is conducted on three small graph datasets. Results on more larger datasets [3] are expected to support the effectiveness of the proposed method.

Reference:
[1] Contrastive Multi-View Representation Learning on Graphs, ICML 2020
[2] Big Self-Supervised Models are Strong Semi-Supervised Learners, NeurIPS 2020
[3] Open Graph Benchmark: Datasets for Machine Learning on Graphs

---

> ### Author Response · Authors · 2020-11-15
> **Response to Reviewer 5**
>
> First of all, thank you very much for your valuable comments on our paper. However, we found that there are some discrepancies in your comment with our paper, and we are here to present a statement. What we use in our paper is the consistency between the two views, instead of constructing positive and negative pairs by data enhancement as contrastive learning, that is，our paper is quite different from contrastive learning model. Therefore, the similarity to contrastive learning mentioned in your comment does not hold. As for self-training, our method is indeed used.  We use self-training to extract pseudo-labels to provide sufficient guidance for model training, but our purpose of using self-training is to provide consistency to the model. Regarding the second shortcoming mentioned in your comment, we think you may have misunderstood it. What we express in the paper is that the existing methods require a large number of known label nodes for training, and our proposed method can perform efficiently in the case of label scarcity, but it is not effective on large graphs as you said.

---

### Decision · Program_Chairs · 2021-01-07
**Final Decision**

**Decision:**

Reject

**Comment:**

The authors consider view-consistency when learning graph neural networks. However, as mentioned by the reviewers, the novelty of the proposed method is limited and the rationality of the implementation is not convincing. More deep discussions about related papers and analytic experiments are required to support this work. Additionally, I have concerns about the scalability of the method --- whether it can deal with more than two views and how it will perform are not studied in this work. I tend to reject it based on its current status.